# #Exploration: A Study of Count-Based Exploration for Deep Reinforcement Learning

**Haoran Tang**[1*], **Rein Houthooft**[34*], **Davis Foote**[2], **Adam Stooke**[2], **Xi Chen**[2†],
**Yan Duan**[2†], **John Schulman**[4], **Filip De Turck**[3], **Pieter Abbeel** [2†]

[1] UC Berkeley, Department of Mathematics
[2] UC Berkeley, Department of Electrical Engineering and Computer Sciences
[3] Ghent University – imec, Department of Information Technology
[4] OpenAI

## Abstract

Count-based exploration algorithms are known to perform near-optimally when used in conjunction with tabular reinforcement learning (RL) methods for solving small discrete Markov decision processes (MDPs). It is generally thought that count-based methods cannot be applied in high-dimensional state spaces, since most states will only occur once. Recent deep RL exploration strategies are able to deal with high-dimensional continuous state spaces through complex heuristics, often relying on *optimism in the face of uncertainty* or *intrinsic motivation*. In this work, we describe a surprising finding: a simple generalization of the classic count-based approach can reach near state-of-the-art performance on various high-dimensional and/or continuous deep RL benchmarks. States are mapped to hash codes, which allows to count their occurrences with a hash table. These counts are then used to compute a reward bonus according to the classic count-based exploration theory. We find that simple hash functions can achieve surprisingly good results on many challenging tasks. Furthermore, we show that a domain-dependent learned hash code may further improve these results. Detailed analysis reveals important aspects of a good hash function: 1) having appropriate granularity and 2) encoding information relevant to solving the MDP. This exploration strategy achieves near state-of-the-art performance on both continuous control tasks and Atari 2600 games, hence providing a simple yet powerful baseline for solving MDPs that require considerable exploration.

## 1 Introduction

Reinforcement learning (RL) studies an agent acting in an initially unknown environment, learning through trial and error to maximize rewards. It is impossible for the agent to act near-optimally until it has sufficiently explored the environment and identified all of the opportunities for high reward, in all scenarios. A core challenge in RL is how to balance exploration—actively seeking out novel states and actions that might yield high rewards and lead to long-term gains; and exploitation—maximizing short-term rewards using the agent's current knowledge. While there are exploration techniques for finite MDPs that enjoy theoretical guarantees, there are no fully satisfying techniques for high-dimensional state spaces; therefore, developing more general and robust exploration techniques is an active area of research.

---

[*]These authors contributed equally. Correspondence to: Haoran Tang <hrtang@math.berkeley.edu>, Rein Houthooft <rein.houthooft@openai.com>

[†]Work done at OpenAI

Most of the recent state-of-the-art RL results have been obtained using simple exploration strategies such as uniform sampling [21] and i.i.d./correlated Gaussian noise [19, 30]. Although these heuristics are sufficient in tasks with well-shaped rewards, the sample complexity can grow exponentially (with state space size) in tasks with sparse rewards [25]. Recently developed exploration strategies for deep RL have led to significantly improved performance on environments with sparse rewards. Bootstrapped DQN [24] led to faster learning in a range of Atari 2600 games by training an ensemble of Q-functions. Intrinsic motivation methods using pseudo-counts achieve state-of-the-art performance on Montezuma's Revenge, an extremely challenging Atari 2600 game [4]. Variational Information Maximizing Exploration (VIME, [13]) encourages the agent to explore by acquiring information about environment dynamics, and performs well on various robotic locomotion problems with sparse rewards. However, we have not seen a very simple and fast method that can work across different domains.

Some of the classic, theoretically-justified exploration methods are based on counting state-action visitations, and turning this count into a bonus reward. In the bandit setting, the well-known UCB algorithm of [18] chooses the action $a_t$ at time $t$ that maximizes $\hat{r}(a_t) + \sqrt{\frac{2 \log t}{n(a_t)}}$ where $\hat{r}(a_t)$ is the estimated reward, and $n(a_t)$ is the number of times action $a_t$ was previously chosen. In the MDP setting, some of the algorithms have similar structure, for example, Model Based Interval Estimation–Exploration Bonus (MBIE-EB) of [34] counts state-action pairs with a table $n(s, a)$ and adding a bonus reward of the form $\frac{\beta}{\sqrt{n(s,a)}}$ to encourage exploring less visited pairs. [16] show that the inverse-square-root dependence is optimal. MBIE and related algorithms assume that the augmented MDP is solved analytically at each timestep, which is only practical for small finite state spaces.

This paper presents a simple approach for exploration, which extends classic counting-based methods to high-dimensional, continuous state spaces. We discretize the state space with a hash function and apply a bonus based on the state-visitation count. The hash function can be chosen to appropriately balance generalization across states, and distinguishing between states. We select problems from rllab [8] and Atari 2600 [3] featuring sparse rewards, and demonstrate near state-of-the-art performance on several games known to be hard for naïve exploration strategies. The main strength of the presented approach is that it is fast, flexible and complementary to most existing RL algorithms.

In summary, this paper proposes a generalization of classic count-based exploration to high-dimensional spaces through hashing (Section 2); demonstrates its effectiveness on challenging deep RL benchmark problems and analyzes key components of well-designed hash functions (Section 4).

## 2 Methodology

### 2.1 Notation

This paper assumes a finite-horizon discounted Markov decision process (MDP), defined by $(\mathcal{S}, \mathcal{A}, \mathcal{P}, r, \rho_0, \gamma, T)$, in which $\mathcal{S}$ is the state space, $\mathcal{A}$ the action space, $\mathcal{P}$ a transition probability distribution, $r : \mathcal{S} \times \mathcal{A} \to \mathbb{R}$ a reward function, $\rho_0$ an initial state distribution, $\gamma \in (0, 1]$ a discount factor, and $T$ the horizon. The goal of RL is to maximize the total expected discounted reward $\mathbb{E}_{\pi, \mathcal{P}} \left[ \sum_{t=0}^{T} \gamma^t r(s_t, a_t) \right]$ over a policy $\pi$, which outputs a distribution over actions given a state.

### 2.2 Count-Based Exploration via Static Hashing

Our approach discretizes the state space with a hash function $\phi : \mathcal{S} \to \mathbb{Z}$. An exploration bonus $r^+ : \mathcal{S} \to \mathbb{R}$ is added to the reward function, defined as

$$r^+(s) = \frac{\beta}{\sqrt{n(\phi(s))}}, \tag{1}$$

where $\beta \in \mathbb{R}_{\geq 0}$ is the bonus coefficient. Initially the counts $n(\cdot)$ are set to zero for the whole range of $\phi$. For every state $s_t$ encountered at time step $t$, $n(\phi(s_t))$ is increased by one. The agent is trained with rewards $(r + r^+)$, while performance is evaluated as the sum of rewards without bonuses.

**Algorithm 1:** Count-based exploration through static hashing, using SimHash

**1** Define state preprocessor $g : \mathcal{S} \to \mathbb{R}^D$

**2** (In case of SimHash) Initialize $A \in \mathbb{R}^{k \times D}$ with entries drawn i.i.d. from the standard Gaussian distribution $\mathcal{N}(0, 1)$

**3** Initialize a hash table with values $n(\cdot) \equiv 0$

**4 for** each iteration $j$ **do**

**5** &emsp; Collect a set of state-action samples $\{(s_m, a_m)\}_{m=0}^{M}$ with policy $\pi$

**6** &emsp; Compute hash codes through any LSH method, e.g., for SimHash, $\phi(s_m) = \mathrm{sgn}(Ag(s_m))$

**7** &emsp; Update the hash table counts $\forall m : 0 \le m \le M$ as $n(\phi(s_m)) \leftarrow n(\phi(s_m)) + 1$

**8** &emsp; Update the policy $\pi$ using rewards $\left\{ r(s_m, a_m) + \dfrac{\beta}{\sqrt{n(\phi(s_m))}} \right\}_{m=0}^{M}$ with any RL algorithm

Note that our approach is a departure from count-based exploration methods such as MBIE-EB since we use a state-space count $n(s)$ rather than a state-action count $n(s, a)$. State-action counts $n(s, a)$ are investigated in the Supplementary Material, but no significant performance gains over state counting could be witnessed. A possible reason is that the policy itself is sufficiently random to try most actions at a novel state.

Clearly the performance of this method will strongly depend on the choice of hash function $\phi$. One important choice we can make regards the *granularity* of the discretization: we would like for "distant" states to be be counted separately while "similar" states are merged. If desired, we can incorporate prior knowledge into the choice of $\phi$, if there would be a set of salient state features which are known to be relevant. A short discussion on this matter is given in the Supplementary Material.

Algorithm 1 summarizes our method. The main idea is to use locality-sensitive hashing (LSH) to convert continuous, high-dimensional data to discrete hash codes. LSH is a popular class of hash functions for querying nearest neighbors based on certain similarity metrics [2]. A computationally efficient type of LSH is SimHash [6], which measures similarity by angular distance. SimHash retrieves a binary code of state $s \in \mathcal{S}$ as

$$\phi(s) = \mathrm{sgn}(Ag(s)) \in \{-1, 1\}^k, \tag{2}$$

where $g : \mathcal{S} \to \mathbb{R}^D$ is an optional preprocessing function and $A$ is a $k \times D$ matrix with i.i.d. entries drawn from a standard Gaussian distribution $\mathcal{N}(0, 1)$. The value for $k$ controls the granularity: higher values lead to fewer collisions and are thus more likely to distinguish states.

## 2.3 Count-Based Exploration via Learned Hashing

When the MDP states have a complex structure, as is the case with image observations, measuring their similarity directly in pixel space fails to provide the semantic similarity measure one would desire. Previous work in computer vision [7, 20, 36] introduce manually designed feature representations of images that are suitable for semantic tasks including detection and classification. More recent methods learn complex features directly from data by training convolutional neural networks [12, 17, 31]. Considering these results, it may be difficult for a method such as SimHash to cluster states appropriately using only raw pixels.

Therefore, rather than using SimHash, we propose to use an autoencoder (AE) to learn meaningful hash codes in one of its hidden layers as a more advanced LSH method. This AE takes as input states $s$ and contains one special dense layer comprised of $D$ sigmoid functions. By rounding the sigmoid activations $b(s)$ of this layer to their closest binary number $\lfloor b(s) \rceil \in \{0, 1\}^D$, any state $s$ can be binarized. This is illustrated in Figure 1 for a convolutional AE.

A problem with this architecture is that dissimilar inputs $s_i, s_j$ can map to identical hash codes $\lfloor b(s_i) \rceil = \lfloor b(s_j) \rceil$, but the AE still reconstructs them perfectly. For example, if $b(s_i)$ and $b(s_j)$ have values 0.6 and 0.7 at a particular dimension, the difference can be exploited by deconvolutional layers in order to reconstruct $s_i$ and $s_j$ perfectly, although that dimension rounds to the same binary value. One can imagine replacing the bottleneck layer $b(s)$ with the hash codes $\lfloor b(s) \rceil$, but then gradients cannot be back-propagated through the rounding function. A solution is proposed by Gregor et al. [10] and Salakhutdinov & Hinton [28] is to inject uniform noise $U(-a, a)$ into the sigmoid

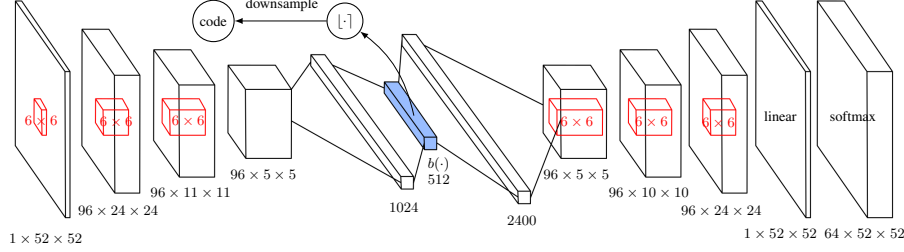

Figure 1: The autoencoder (AE) architecture for ALE; the solid block represents the dense sigmoidal binary code layer, after which noise $U(-a, a)$ is injected.

---

**Algorithm 2:** Count-based exploration using learned hash codes

---

1   Define state preprocessor $g : \mathcal{S} \to \{0, 1\}^D$ as the binary code resulting from the autoencoder (AE)

2   Initialize $A \in \mathbb{R}^{k \times D}$ with entries drawn i.i.d. from the standard Gaussian distribution $\mathcal{N}(0, 1)$

3   Initialize a hash table with values $n(\cdot) \equiv 0$

4   **for** each iteration $j$ **do**

5      Collect a set of state-action samples $\{(s_m, a_m)\}_{m=0}^M$ with policy $\pi$

6      Add the state samples $\{s_m\}_{m=0}^M$ to a FIFO replay pool $\mathcal{R}$

7      **if** $j \bmod j_{\text{update}} = 0$ **then**

8         Update the AE loss function in Eq. (3) using samples drawn from the replay pool $\{s_n\}_{n=1}^N \sim \mathcal{R}$, for example using stochastic gradient descent

9      Compute $g(s_m) = \lfloor b(s_m) \rceil$, the $D$-dim rounded hash code for $s_m$ learned by the AE

10      Project $g(s_m)$ to a lower dimension $k$ via SimHash as $\phi(s_m) = \text{sgn}(Ag(s_m))$

11      Update the hash table counts $\forall m : 0 \le m \le M$ as $n(\phi(s_m)) \leftarrow n(\phi(s_m)) + 1$

12      Update the policy $\pi$ using rewards $\left\{ r(s_m, a_m) + \frac{\beta}{\sqrt{n(\phi(s_m))}} \right\}_{m=0}^M$ with any RL algorithm

---

activations. By choosing uniform noise with $a > \frac{1}{4}$, the AE is only capable of (always) reconstructing distinct state inputs $s_i \ne s_j$, if it has learned to spread the sigmoid outputs sufficiently far apart, $|b(s_i) - b(s_j)| > \epsilon$, in order to counteract the injected noise.

As such, the loss function over a set of collected states $\{s_i\}_{i=1}^N$ is defined as

$$L\left(\{s_n\}_{n=1}^N\right) = -\frac{1}{N} \sum_{n=1}^N \left[ \log p(s_n) - \frac{\lambda}{K} \sum_{i=1}^D \min\left\{ (1 - b_i(s_n))^2, b_i(s_n)^2 \right\} \right], \quad (3)$$

with $p(s_n)$ the AE output. This objective function consists of a negative log-likelihood term and a term that pressures the binary code layer to take on binary values, scaled by $\lambda \in \mathbb{R}_{\ge 0}$. The reasoning behind this latter term is that it might happen that for particular states, a certain sigmoid unit is never used. Therefore, its value might fluctuate around $\frac{1}{2}$, causing the corresponding bit in binary code $\lfloor b(s) \rceil$ to flip over the agent lifetime. Adding this second loss term ensures that an unused bit takes on an arbitrary binary value.

For Atari 2600 image inputs, since the pixel intensities are discrete values in the range $[0, 255]$, we make use of a pixel-wise softmax output layer [37] that shares weights between all pixels. The architectural details are described in the Supplementary Material and are depicted in Figure 1. Because the code dimension often needs to be large in order to correctly reconstruct the input, we apply a downsampling procedure to the resulting binary code $\lfloor b(s) \rceil$, which can be done through random projection to a lower-dimensional space via SimHash as in Eq. (2).

On the one hand, it is important that the mapping from state to code needs to remain relatively consistent over time, which is nontrivial as the AE is constantly updated according to the latest data (Algorithm 2 line 8). A solution is to downsample the binary code to a very low dimension, or by slowing down the training process. On the other hand, the code has to remain relatively unique

for states that are both distinct and close together on the image manifold. This is tackled both by the second term in Eq. (3) and by the saturating behavior of the sigmoid units. States already well represented by the AE tend to saturate the sigmoid activations, causing the resulting loss gradients to be close to zero, making the code less prone to change.

# 3 Related Work

Classic count-based methods such as MBIE [33], MBIE-EB and [16] solve an approximate Bellman equation as an inner loop before the agent takes an action [34]. As such, bonus rewards are propagated immediately throughout the state-action space. In contrast, contemporary deep RL algorithms propagate the bonus signal based on rollouts collected from interacting with environments, with value-based [21] or policy gradient-based [22, 30] methods, at limited speed. In addition, our proposed method is intended to work with contemporary deep RL algorithms, it differs from classical count-based method in that our method relies on visiting unseen states first, before the bonus reward can be assigned, making uninformed exploration strategies still a necessity at the beginning. Filling the gaps between our method and classic theories is an important direction of future research.

A related line of classical exploration methods is based on the idea of *optimism in the face of uncertainty* [5] but not restricted to using counting to implement "optimism", e.g., R-Max [5], UCRL [14], and $E^3$ [15]. These methods, similar to MBIE and MBIE-EB, have theoretical guarantees in tabular settings.

Bayesian RL methods [9, 11, 16, 35], which keep track of a distribution over MDPs, are an alternative to optimism-based methods. Extensions to continuous state space have been proposed by [27] and [25].

Another type of exploration is curiosity-based exploration. These methods try to capture the agent's surprise about transition dynamics. As the agent tries to optimize for surprise, it naturally discovers novel states. We refer the reader to [29] and [26] for an extensive review on curiosity and intrinsic rewards.

Several exploration strategies for deep RL have been proposed to handle high-dimensional state space recently. [13] propose VIME, in which information gain is measured in Bayesian neural networks modeling the MDP dynamics, which is used an exploration bonus. [32] propose to use the prediction error of a learned dynamics model as an exploration bonus. Thompson sampling through bootstrapping is proposed by [24], using bootstrapped Q-functions.

The most related exploration strategy is proposed by [4], in which an exploration bonus is added inversely proportional to the square root of a *pseudo-count* quantity. A state pseudo-count is derived from its log-probability improvement according to a density model over the state space, which in the limit converges to the empirical count. Our method is similar to pseudo-count approach in the sense that both methods are performing approximate counting to have the necessary generalization over unseen states. The difference is that a density model has to be designed and learned to achieve good generalization for pseudo-count whereas in our case generalization is obtained by a wide range of simple hash functions (not necessarily SimHash). Another interesting connection is that our method also implies a density model $\rho(s) = \frac{n(\phi(s))}{N}$ over all visited states, where $N$ is the total number of states visited. Another method similar to hashing is proposed by [1], which clusters states and counts cluster centers instead of the true states, but this method has yet to be tested on standard exploration benchmark problems.

# 4 Experiments

Experiments were designed to investigate and answer the following research questions:

1. Can count-based exploration through hashing improve performance significantly across different domains? How does the proposed method compare to the current state of the art in exploration for deep RL?

2. What is the impact of learned or static state preprocessing on the overall performance when image observations are used?

To answer question 1, we run the proposed method on deep RL benchmarks (rllab and ALE) that feature sparse rewards, and compare it to other state-of-the-art algorithms. Question 2 is answered by trying out different image preprocessors on Atari 2600 games. Trust Region Policy Optimization (TRPO, [30]) is chosen as the RL algorithm for all experiments, because it can handle both discrete and continuous action spaces, can conveniently ensure stable improvement in the policy performance, and is relatively insensitive to hyperparameter changes. The hyperparameters settings are reported in the Supplementary Material.

## 4.1 Continuous Control

The rllab benchmark [8] consists of various control tasks to test deep RL algorithms. We selected several variants of the basic and locomotion tasks that use sparse rewards, as shown in Figure 2, and adopt the experimental setup as defined in [13]—a description can be found in the Supplementary Material. These tasks are all highly difficult to solve with naïve exploration strategies, such as adding Gaussian noise to the actions.

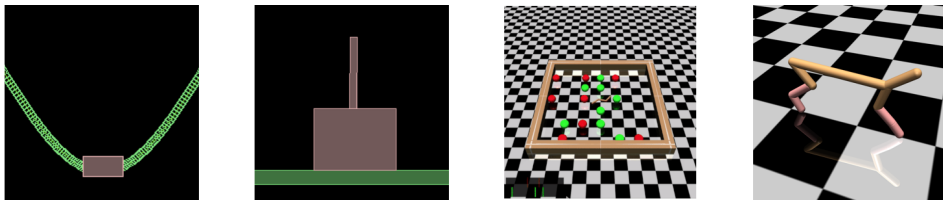

Figure 2: Illustrations of the rllab tasks used in the continuous control experiments, namely MountainCar, CartPoleSwingup, SimmerGather, and HalfCheetah; taken from [8].

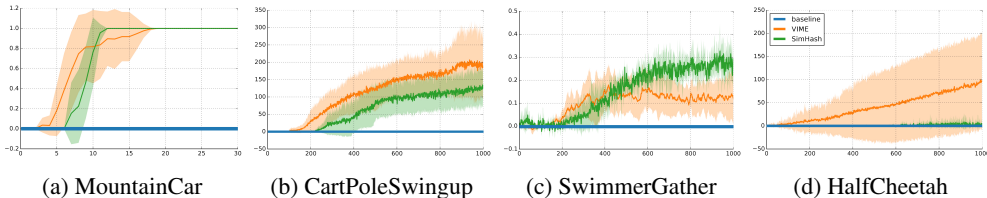

(a) MountainCar     (b) CartPoleSwingup     (c) SwimmerGather     (d) HalfCheetah

Figure 3: Mean average return of different algorithms on rllab tasks with sparse rewards. The solid line represents the mean average return, while the shaded area represents one standard deviation, over 5 seeds for the baseline and SimHash (the baseline curves happen to overlap with the axis).

Figure 3 shows the results of TRPO (baseline), TRPO-SimHash, and VIME [13] on the classic tasks MountainCar and CartPoleSwingup, the locomotion task HalfCheetah, and the hierarchical task SwimmerGather. Using count-based exploration with hashing is capable of reaching the goal in all environments (which corresponds to a nonzero return), while baseline TRPO with Gaussia n control noise fails completely. Although TRPO-SimHash picks up the sparse reward on HalfCheetah, it does not perform as well as VIME. In contrast, the performance of SimHash is comparable with VIME on MountainCar, while it outperforms VIME on SwimmerGather.

## 4.2 Arcade Learning Environment

The Arcade Learning Environment (ALE, [3]), which consists of Atari 2600 video games, is an important benchmark for deep RL due to its high-dimensional state space and wide variety of games. In order to demonstrate the effectiveness of the proposed exploration strategy, six games are selected featuring long horizons while requiring significant exploration: Freeway, Frostbite, Gravitar, Montezuma's Revenge, Solaris, and Venture. The agent is trained for $500$ iterations in all experiments, with each iteration consisting of $0.1\,\mathrm{M}$ steps (the TRPO batch size, corresponds to $0.4\,\mathrm{M}$ frames). Policies and value functions are neural networks with identical architectures to [22]. Although the policy and baseline take into account the previous four frames, the counting algorithm only looks at the latest frame.

Table 1: Atari 2600: average total reward after training for $50\,\mathrm{M}$ time steps. Boldface numbers indicate best results. Italic numbers are the best among our methods.

|  | Freeway | Frostbite | Gravitar | Montezuma | Solaris | Venture |
|---|---|---|---|---|---|---|
| TRPO (baseline) | 16.5 | 2869 | 486 | 0 | 2758 | 121 |
| TRPO-pixel-SimHash | 31.6 | 4683 | 468 | 0 | 2897 | 263 |
| TRPO-BASS-SimHash | 28.4 | 3150 | *604* | *238* | 1201 | *616* |
| TRPO-AE-SimHash | ***33.5*** | ***5214*** | 482 | 75 | *4467* | 445 |
| Double-DQN | 33.3 | 1683 | 412 | 0 | 3068 | 98.0 |
| Dueling network | 0.0 | 4672 | 588 | 0 | 2251 | 497 |
| Gorila | 11.7 | 605 | **1054** | 4 | N/A | **1245** |
| DQN Pop-Art | 33.4 | 3469 | 483 | 0 | **4544** | 1172 |
| A3C+ | 27.3 | 507 | 246 | 142 | 2175 | 0 |
| pseudo-count | 29.2 | 1450 | – | **3439** | – | 369 |

**BASS**  To compare with the autoencoder-based learned hash code, we propose using Basic Abstraction of the ScreenShots (BASS, also called Basic; see [3]) as a static preprocessing function $g$. BASS is a hand-designed feature transformation for images in Atari 2600 games. BASS builds on the following observations specific to Atari: 1) the game screen has a low resolution, 2) most objects are large and monochrome, and 3) winning depends mostly on knowing object locations and motions. We designed an adapted version of BASS[3], that divides the RGB screen into square cells, computes the average intensity of each color channel inside a cell, and assigns the resulting values to bins that uniformly partition the intensity range $[0, 255]$. Mathematically, let $C$ be the cell size (width and height), $B$ the number of bins, $(i, j)$ cell location, $(x, y)$ pixel location, and $z$ the channel, then

$$\text{feature}(i, j, z) = \left\lfloor \frac{B}{255C^2} \sum_{(x,y)\in\,\text{cell}(i,j)} I(x, y, z) \right\rfloor. \tag{4}$$

Afterwards, the resulting integer-valued feature tensor is converted to an integer hash code ($\phi(s_t)$ in Line 6 of Algorithm 1). A BASS feature can be regarded as a miniature that efficiently encodes object locations, but remains invariant to negligible object motions. It is easy to implement and introduces little computation overhead. However, it is designed for generic Atari game images and may not capture the structure of each specific game very well.

We compare our results to double DQN [39], dueling network [40], A3C+ [4], double DQN with pseudo-counts [4], Gorila [23], and DQN Pop-Art [38] on the "null op" metric[4]. We show training curves in Figure 4 and summarize all results in Table 1. Surprisingly, TRPO-pixel-SimHash already outperforms the baseline by a large margin and beats the previous best result on Frostbite. TRPO-BASS-SimHash achieves significant improvement over TRPO-pixel-SimHash on Montezuma's Revenge and Venture, where it captures object locations better than other methods.[5] TRPO-AE-SimHash achieves near state-of-the-art performance on Freeway, Frostbite and Solaris.

As observed in Table 1, preprocessing images with BASS or using a learned hash code through the AE leads to much better performance on Gravitar, Montezuma's Revenge and Venture. Therefore, a static or adaptive preprocessing step can be important for a good hash function.

In conclusion, our count-based exploration method is able to achieve remarkable performance gains even with simple hash functions like SimHash on the raw pixel space. If coupled with domain-dependent state preprocessing techniques, it can sometimes achieve far better results.

A reason why our proposed method does not achieve state-of-the-art performance on all games is that TRPO does not reuse off-policy experience, in contrast to DQN-based algorithms [4, 23, 38]), and is

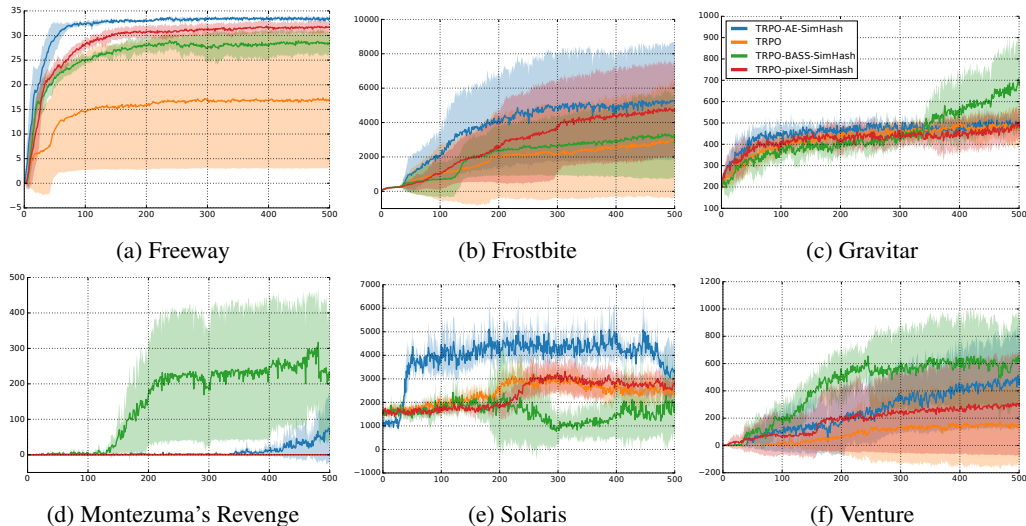

(a) Freeway        (b) Frostbite        (c) Gravitar

(d) Montezuma's Revenge      (e) Solaris        (f) Venture

Figure 4: Atari 2600 games: the solid line is the mean average undiscounted return per iteration, while the shaded areas represent the one standard deviation, over 5 seeds for the baseline, TRPO-pixel-SimHash, and TRPO-BASS-SimHash, while over 3 seeds for TRPO-AE-SimHash.

hence less efficient in harnessing extremely sparse rewards. This explanation is corroborated by the experiments done in [4], in which A3C+ (an on-policy algorithm) scores much lower than DQN (an off-policy algorithm), while using the exact same exploration bonus.

## 5   Conclusions

This paper demonstrates that a generalization of classical counting techniques through hashing is able to provide an appropriate signal for exploration, even in continuous and/or high-dimensional MDPs using function approximators, resulting in near state-of-the-art performance across benchmarks. It provides a simple yet powerful baseline for solving MDPs that require informed exploration.

## Acknowledgments

We would like to thank our colleagues at Berkeley and OpenAI for insightful discussions. This research was funded in part by ONR through a PECASE award. Yan Duan was also supported by a Berkeley AI Research lab Fellowship and a Huawei Fellowship. Xi Chen was also supported by a Berkeley AI Research lab Fellowship. We gratefully acknowledge the support of the NSF through grant IIS-1619362 and of the ARC through a Laureate Fellowship (FL110100281) and through the ARC Centre of Excellence for Mathematical and Statistical Frontiers. Adam Stooke gratefully acknowledges funding from a Fannie and John Hertz Foundation fellowship. Rein Houthooft was supported by a Ph.D. Fellowship of the Research Foundation - Flanders (FWO).

## Footnotes

[3]The original BASS exploits the fact that at most 128 colors can appear on the screen. Our adapted version does not make this assumption.

[4]The agent takes no action for a random number (within 30) of frames at the beginning of each episode.

[5]We provide videos of example game play and visualizations of the difference bewteen Pixel-SimHash and BASS-SimHash at https://www.youtube.com/playlist?list=PLAd-UMX6FkBQdLNWtY8nH1-pzYJA_1T55

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
