[Supplementary Material]

# Supplementary Material

## 1 Hyperparameter Settings

Throughout all experiments, we use Adam [8] for optimizing the baseline function and the autoencoder. Hyperparameters for rllab experiments are summarized in Table 1. Here the policy takes a state $s$ as input, and outputs a Gaussian distribution $\mathcal{N}(\mu(s), \sigma^2)$, where $\mu(s)$ is the output of a multi-layer perceptron (MLP) with tanh nonlinearity, and $\sigma > 0$ is a state-independent parameter.

Table 1: TRPO hyperparameters for rllab experiments

| Experiment | MountainCar | CartPoleSwingUp | HalfCheetah | SwimmerGatherer |
|---|---|---|---|---|
| TRPO batch size | 5k | 5k | 5k | 50k |
| TRPO step size | | 0.01 | | |
| Discount factor $\gamma$ | | 0.99 | | |
| Policy hidden units | (32, 32) | (32, ) | (32, 32) | (64, 32) |
| Baseline function | linear | linear | linear | MLP: 32 units |
| Exploration bonus | | $\beta = 0.01$ | | |
| SimHash dimension | | $k = 32$ | | |

Hyperparameters for Atari 2600 experiments are summarized in Table 2 and 3. By default, all convolutional layers are followed by ReLU nonlinearity.

Table 2: TRPO hyperparameters for Atari experiments with image input

| Experiment | TRPO-pixel-SimHash | TRPO-BASS-SimHash | TRPO-AE-SimHash |
|---|---|---|---|
| TRPO batch size | | 100k | |
| TRPO step size | | 0.01 | |
| Discount factor | | 0.995 | |
| # random seeds | 5 | 5 | 3 |
| Input preprocessing | grayscale; downsampled to $52 \times 52$; each pixel rescaled to $[-1, 1]$ | | |
| | 4 previous frames are concatenated to form the input state | | |
| Policy structure | 16 conv filters of size $8 \times 8$, stride 4 | | |
| | 32 conv filters of size $4 \times 4$, stride 2 | | |
| | fully-connect layer with 256 units | | |
| | linear transform and softmax to output action probabilities | | |
| | (use batch normalization[7] at every layer) | | |
| Baseline structure | (same as policy, except that the last layer is a single scalar) | | |
| Exploration bonus | | $\beta = 0.01$ | |
| Hashing parameters | $k = 256$ | cell size $C = 20$ | $b(s)$ size: 256 bits |
| | | $B = 20$ bins | downsampled to 64 bits |

The autoencoder architecture was shown in Figure 1 of Section 2.3. Specifically, uniform noise $U(-a, a)$ with $a = 0.3$ is added to the sigmoid activations. The loss function Eq.(3) (in the main

Table 3: TRPO hyperparameters for Atari experiments with RAM input

| Experiment | TRPO-RAM-SimHash |
|---|---|
| TRPO batch size | 100k |
| TRPO step size | 0.01 |
| Discount factor | 0.995 |
| # random seeds | 10 |
| Input preprocessing | vector of length 128 in the range $[0, 255]$; downsampled to $[-1, 1]$ |
| Policy structure | MLP: (32, 32, number_of_actions), $\tanh$ |
| Baseline structure | MLP: (32, 32, 1), $\tanh$ |
| Exploration bonus | $\beta = 0.01$ |
| SimHash dimension | $k = 256$ |

text), using $\lambda = 10$, is updated every $j_{\text{update}} = 3$ iterations. The architecture looks as follows: an input layer of size $52 \times 52$, representing the image luminance is followed by 3 consecutive $6 \times 6$ convolutional layers with stride 2 and 96 filters feed into a fully connected layer of size 1024, which connects to the binary code layer. This binary code layer feeds into a fully-connected layer of 1024 units, connecting to a fully-connected layer of 2400 units. This layer feeds into 3 consecutive $6 \times 6$ transposed convolutional layers of which the final one connects to a pixel-wise softmax layer with 64 bins, representing the pixel intensities. Moreover, label smoothing is applied to the different softmax bins, in which the log-probability of each of the bins is increased by 0.003, before normalizing. The softmax weights are shared among each pixel.

In addition, we apply counting Bloom filters [5] to maintain a small hash table. Details can be found in Appendix 4.

## 2   Description of the Adapted rllab Tasks

This section describes the continuous control environments used in the experiments. The tasks are implemented as described in [4], following the sparse reward adaptation of [6]. The tasks have the following state and action dimensions: CartPoleSwingup, $\mathcal{S} \subseteq \mathbb{R}^4$, $\mathcal{A} \subseteq \mathbb{R}$; MountainCar $\mathcal{S} \subseteq \mathbb{R}^3$, $\mathcal{A} \subseteq \mathbb{R}^1$; HalfCheetah, $\mathcal{S} \subseteq \mathbb{R}^{20}$, $\mathcal{A} \subseteq \mathbb{R}^6$; SwimmerGather, $\mathcal{S} \subseteq \mathbb{R}^{33}$, $\mathcal{A} \subseteq \mathbb{R}^2$. For the sparse reward experiments, the tasks have been modified as follows. In CartPoleSwingup, the agent receives a reward of $+1$ when $\cos(\beta) > 0.8$, with $\beta$ the pole angle. In MountainCar, the agent receives a reward of $+1$ when the goal state is reached, namely escaping the valley from the right side. Therefore, the agent has to figure out how to swing up the pole in the absence of any initial external rewards. In HalfCheetah, the agent receives a reward of $+1$ when $x_{\text{body}} > 5$. As such, it has to figure out how to move forward without any initial external reward. The time horizon is set to $T = 500$ for all tasks.

## 3   Analysis of Learned Binary Representation

Figure 1 shows the downsampled codes learned by the autoencoder for several Atari 2600 games (Frostbite, Freeway, and Montezuma's Revenge). Each row depicts 50 consecutive frames (from 0 to 49, going from left to right, top to bottom). The pictures in the right column depict the binary codes that correspond with each of these frames (one frame per row). Figure 2 shows the reconstructions of several subsequent images according to the autoencoder. Some binaries stay consistent across frames, and some appear to respond to specific objects or events. Although the precise meaning of each binary number is not immediately obvious, the figure suggests that the learned hash code is a reasonable abstraction of the game state.

## 4   Counting Bloom Filter/Count-Min Sketch

We experimented with directly building a hashing dictionary with keys $\phi(s)$ and values the state counts, but observed an unnecessary increase in computation time. Our implementation converts the integer hash codes into binary numbers and then into the "bytes" type in Python. The hash table is a dictionary using those bytes as keys.

Figure 1: Frostbite, Freeway, and Montezuma's Revenge: subsequent frames (left) and corresponding code (right); the frames are ordered from left (starting with frame number 0) to right, top to bottom; the vertical axis in the right images correspond to the frame number.

47  However, an alternative technique called Count-Min Sketch [3], with a data structure identical
48  to counting Bloom filters [5], can count with a fixed integer array and thus reduce computation
49  time. Specifically, let $p^1, \ldots, p^l$ be distinct large prime numbers and define $\phi^j(s) = \phi(s) \bmod p^j$.
50  The count of state $s$ is returned as $\min_{1 \le j \le l} n^j\big(\phi^j(s)\big)$. To increase the count of $s$, we increment
51  $n^j\big(\phi^j(s)\big)$ by 1 for all $j$. Intuitively, the method replaces $\phi$ by weaker hash functions, while it reduces
52  the probability of over-counting by reporting counts agreed by all such weaker hash functions. The
53  final hash code is represented as $\big(\phi^1(s), \ldots, \phi^l(s)\big)$.

54  Throughout all experiments above, the prime numbers for the counting Bloom filter are 999931,
55  999953, 999959, 999961, 999979, and 999983, which we abbreviate as "6 M". In addition, we
56  experimented with 6 other prime numbers, each approximately 15 M, which we abbreviate as "90 M".
57  As we can see in Figure 3, counting states with a dictionary or with Bloom filters lead to similar
58  performance, but the computation time of latter is lower. Moreover, there is little difference between
59  direct counting and using a very larger table for Bloom filters, as the average bonus rewards are
60  almost the same, indicating the same degree of exploration-exploitation trade-off. On the other hand,
61  Bloom filters require a fixed table size, which may not be known beforehand.

62  **Theory of Bloom Filters**   Bloom filters [2] are popular for determining whether a data sample $s'$
63  belongs to a dataset $\mathcal{D}$. Suppose we have $l$ functions $\phi^j$ that independently assign each data sample
64  to an integer between 1 and $p$ uniformly at random. Initially $1, 2, \ldots, p$ are marked as 0. Then every
65  $s \in \mathcal{D}$ is "inserted" through marking $\phi^j(s)$ as 1 for all $j$. A new sample $s'$ is reported as a member
66  of $\mathcal{D}$ only if $\phi^j(s)$ are marked as 1 for all $j$. A bloom filter has zero false negative rate (any $s \in \mathcal{D}$ is
67  reported a member), while the false positive rate (probability of reporting a nonmember as a member)
68  decays exponentially in $l$.

69  Though Bloom filters support data insertion, it does not allow data deletion. Counting Bloom filters
70  [5] maintain a counter $n(\cdot)$ for each number between 1 and $p$. Inserting/deleting $s$ corresponds
71  to incrementing/decrementing $n\big(\phi^j(s)\big)$ by 1 for all $j$. Similarly, $s$ is considered a member if
72  $\forall j : n\big(\phi^j(s)\big) = 0$.

73  Count-Min sketch is designed to support memory-efficient counting without introducing too many
74  over-counts. It maintains a separate count $n^j$ for each hash function $\phi^j$ defined as $\phi^j(s) = \phi(s)$
75  $\bmod p^j$, where $p^j$ is a large prime number. For simplicity, we may assume that $p^j \approx p \; \forall j$ and $\phi^j$
76  assigns $s$ to any of $1, \ldots, p$ with uniform probability.

77  We now derive the probability of over-counting. Let $s$ be a fixed data sample (not necessarily
78  inserted yet) and suppose a dataset $\mathcal{D}$ of $N$ samples are inserted. We assume that $p^l \gg N$. Let
79  $n := \min_{1 \le j \le l} n^j\big(\phi^j(s)\big)$ be the count returned by the Bloom filter. We are interested in computing
80  $\mathrm{Prob}(n > 0 | s \notin \mathcal{D})$. Due to assumptions about $\phi^j$, we know $n^j(\phi(s)) \sim \mathrm{Binomial}\left(N, \frac{1}{p}\right)$.
81  Therefore,

$$
\begin{aligned}
\mathrm{Prob}(n > 0 | s \notin \mathcal{D}) &= \frac{\mathrm{Prob}(n > 0, s \notin \mathcal{D})}{\mathrm{Prob}(s \notin \mathcal{D})} \\
&= \frac{\mathrm{Prob}(n > 0) - \mathrm{Prob}(s \in \mathcal{D})}{\mathrm{Prob}(s \notin \mathcal{D})} \\
&\approx \frac{\mathrm{Prob}(n > 0)}{\mathrm{Prob}(s \notin \mathcal{D})} \\
&= \frac{\prod_{j=1}^{l} \mathrm{Prob}(n^j(\phi^j(s)) > 0)}{(1 - 1/p^l)^N} \\
&= \frac{(1 - (1 - 1/p)^N)^l}{(1 - 1/p^l)^N} \\
&\approx \frac{(1 - e^{-N/p})^l}{e^{-N/p^l}} \\
&\approx (1 - e^{-N/p})^l .
\end{aligned}
\tag{1}
$$

82  In particular, the probability of over-counting decays exponentially in $l$. We refer the readers to [3]
83  for other properties of the Count-Min sketch.

## 5 Robustness Analysis

### 5.1 Granularity

While our proposed method is able to achieve remarkable results without requiring much tuning, the granularity of the hash function should be chosen wisely. Granularity plays a critical role in count-based exploration, where the hash function should cluster states without under-generalizing or over-generalizing. Table 4 summarizes granularity parameters for our hash functions. In Table 5 we summarize the performance of TRPO-pixel-SimHash under different granularities. We choose Frostbite and Venture on which TRPO-pixel-SimHash outperforms the baseline, and choose as reward bonus coefficient $\beta = 0.01 \times \frac{256}{k}$ to keep average bonus rewards at approximately the same scale. $k = 16$ only corresponds to $65536$ distinct hash codes, which is insufficient to distinguish between semantically distinct states and hence leads to worse performance. We observed that $k = 512$ tends to capture trivial image details in Frostbite, leading the agent to believe that every state is new and equally worth exploring. Similar results are observed while tuning the granularity parameters for TRPO-BASS-SimHash and TRPO-AE-SimHash.

Table 4: Granularity parameters of various hash functions

| SimHash | $k$: size of the binary code |
| --- | --- |
| BASS | $C$: cell size |
| | $B$: number of bins for each color channel |
| AE | $k$: downstream SimHash parameter |
| | $\lambda$: binarization parameter |
| SmartHash | $s$: grid size agent $(x, y)$ coordinates |

Table 5: Average score at $50\,\mathrm{M}$ time steps achieved by TRPO-pixel-SimHash

| $k$ | 16 | 64 | 128 | 256 | 512 |
| --- | --- | --- | --- | --- | --- |
| Frostbite | 3326 | 4029 | 3932 | **4683** | 1117 |
| Venture | 0 | 218 | 142 | 263 | **306** |

The best granularity depends on both the hash function and the MDP. While adjusting granularity parameter, we observed that it is important to lower the bonus coefficient as granularity is increased. This is because a higher granularity is likely to cause lower state counts, leading to higher bonus rewards that may overwhelm the true rewards.

Apart from the experimental results shown in Table 1 in the main text and Table 5, additional experiments have been performed to study several properties of our algorithm.

### 5.2 Hyperparameter sensitivity

To study the performance sensitivity to hyperparameter changes, we focus on evaluating TRPO-RAM-SimHash on the Atari 2600 game Frostbite, where the method has a clear advantage over the baseline. Because the final scores can vary between different random seeds, we evaluated each set of hyperparameters with 30 seeds. To reduce computation time and cost, RAM states are used instead of image observations.

The results are summarized in Table 6. Herein, $k$ refers to the length of the binary code for hashing while $\beta$ is the multiplicative coefficient for the reward bonus, as defined in Section 2.2 of the main text. This table demonstrates that most hyperparameter settings outperform the baseline ($\beta = 0$) significantly. Moreover, the final scores show a clear pattern in response to changing hyperparameters. Small $\beta$-values lead to insufficient exploration, while large $\beta$-values cause the bonus rewards to overwhelm the true rewards. With a fixed $k$, the scores are roughly concave in $\beta$, peaking at around 0.2. Higher granularity $k$ leads to better performance. Therefore, it can be concluded that the proposed exploration method is robust to hyperparameter changes in comparison to the baseline, and that the best parameter settings can be obtained from a relatively coarse-grained grid search.

Table 6: TRPO-RAM-SimHash performance robustness to hyperparameter changes on Frostbite

| | | | | | $\beta$ | | | |
|---|---|---|---|---|---|---|---|---|
| $k$ | 0 | 0.01 | 0.05 | 0.1 | 0.2 | 0.4 | 0.8 | 1.6 |
| − | 397 | – | – | – | – | – | – | – |
| 64 | – | 879 | 2464 | 2243 | 2489 | 1587 | 1107 | 441 |
| 128 | – | 1475 | 4248 | 2801 | 3239 | 3621 | 1543 | 395 |
| 256 | – | 2583 | 4497 | 4437 | 7849 | 3516 | 2260 | 374 |

Table 7: Average score at $50\,\mathrm{M}$ time steps achieved by TRPO-SmartHash on Montezuma's Revenge (RAM observations)

| $s$ | 1 | 5 | 10 | 20 | 40 | 60 |
|---|---|---|---|---|---|---|
| score | 2598 | 2500 | **3533** | 3025 | 2500 | 1921 |

Table 8: Interpretation of particular RAM entries in Montezuma's Revenge

| ID | Group | Meaning |
|---|---|---|
| 3 | room | room number |
| 42 | agent | $x$ coordinate |
| 43 | agent | $y$ coordinate |
| 52 | agent | orientation (left/right) |
| 27 | beams | on/off |
| 83 | beams | beam countdown (on: 0, off: $36 \rightarrow 0$) |
| 0 | counter | counts from 0 to 255 and repeats |
| 55 | counter | death scene countdown |
| 67 | objects | Doors, skull, and key in 1st room |
| 47 | skull | $x$ coordinate (1st and 2nd room) |

### 5.3 A Case Study of Montezuma's Revenge

Montezuma's Revenge is widely known for its extremely sparse rewards and difficult exploration [1]. While our method does not outperform [1] on this game, we investigate the reasons behind this through various experiments. The experiment process below again demonstrates the importance of a hash function having the correct granularity and encoding relevant information for solving the MDP.

Our first attempt is to use game RAM states instead of image observations as inputs to the policy, which leads to a game score of 2500 with TRPO-BASS-SimHash. Our second attempt is to manually design a hash function that incorporates domain knowledge, called *SmartHash*, which uses an integer-valued vector consisting of the agent's $(x, y)$ location, room number and other useful RAM information as the hash code. The best SmartHash agent is able to obtain a score of 3500. Still the performance is not optimal. We observe that a slight change in the agent's coordinates does not always result in a semantically distinct state, and thus the hash code may remain unchanged. Therefore we choose grid size $s$ and replace the $x$ coordinate by $\lfloor (x - x_{\min})/s \rfloor$ (similarly for $y$). The bonus coefficient is chosen as $\beta = 0.01\sqrt{s}$ to maintain the scale relative to the true reward[1] (see Table 7). Finally, the best agent is able to obtain 6600 total rewards after training for 1000 iterations ($1000\,\mathrm{M}$ time steps), with a grid size $s = 10$.

Table 9: Performance comparison between state counting (left of the slash) and state-action counting (right of the slash) using TRPO-RAM-SimHash on Frostbite

| $k$ | $\beta$ | | | | | | |
|---|---|---|---|---|---|---|---|
| | 0.01 | 0.05 | 0.1 | 0.2 | 0.4 | 0.8 | 1.6 |
| 64 | 879 / 976 | 2464 / 1491 | 2243 / 3954 | 2489 / 5523 | 1587 / 5985 | 1107 / 2052 | 441 / 742 |
| 128 | 1475 / 808 | 4248 / 4302 | 2801 / 4802 | 3239 / 7291 | 3621 / 4243 | 1543 / 1941 | 395 / 362 |
| 256 | 2583 / 1584 | 4497 / 5402 | 4437 / 5431 | 7849 / 4872 | 3516 / 3175 | 2260 / 1238 | 374 / 96 |

Table 8 lists the semantic interpretation of certain RAM entries in Montezuma's Revenge. SmartHash, as described in Section 5.3, makes use of RAM indices 3, 42, 43, 27, and 67. "Beam walls" are deadly barriers that occur periodically in some rooms.

During our pursuit, we had another interesting discovery that the ideal hash function should not simply cluster states by their visual similarity, but instead by their relevance to solving the MDP. We experimented with including enemy locations in the first two rooms into SmartHash ($s = 10$), and observed that average score dropped to 1672 (at iteration 1000). Though it is important for the agent to dodge enemies, the agent also erroneously "enjoys" watching enemy motions at distance (since new states are constantly observed) and "forgets" that his main objective is to enter other rooms. An alternative hash function keeps the same entry "enemy locations", but instead only puts randomly sampled values in it, which surprisingly achieves better performance (3112). However, by ignoring enemy locations altogether, the agent achieves a much higher score (5661) (see Figure 4). In retrospect, we examine the hash codes generated by BASS-SimHash and find that codes clearly distinguish between visually different states (including various enemy locations), but fails to emphasize that the agent needs to explore different rooms. Again this example showcases the importance of encoding relevant information in designing hash functions.

### 5.4 State and state-action counting

Continuing the results in Table 6, the performance of state-action counting is studied using the same experimental setup, summarized in Table 9. In particular, a bonus reward $r^+(s, a) = \frac{\beta}{\sqrt{n(s,a)}}$ instead of $r^+(s) = \frac{\beta}{\sqrt{n(s)}}$ is assigned. These results show that the relative performance of state counting compared to state-action counting depends highly on the selected hyperparameter settings. However, we notice that the best performance is achieved using state counting with $k = 256$ and $\beta = 0.2$.

## Footnotes

[1]The bonus scaling is chosen by assuming all states are visited uniformly and the average bonus reward should remain the same for any grid size.

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

Figure 2: Freeway: subsequent frames and corresponding code (top); the frames are ordered from left (starting with frame number 0) to right, top to bottom; the vertical axis in the right images correspond to the frame number. Within each image, the left picture is the input frame, the middle picture the reconstruction, and the right picture, the reconstruction error.

(a) Mean average undiscounted return

(b) Average bonus reward

Figure 3: Statistics of TRPO-pixel-SimHash ($k = 256$) on Frostbite. Solid lines are the mean, while the shaded areas represent the one standard deviation. Results are derived from 10 random seeds. Direct counting with a dictionary uses 2.7 times more computations than counting Bloom filters (6 M or 90 M).

Figure 4: SmartHash results on Montezuma's Revenge (RAM observations): the solid line is the mean average undiscounted return per iteration, while the shaded areas represent the one standard deviation, over 5 seeds.