[Reviews · NeurIPS 2017]

Reviewer 1



This paper is already available on arxiv and cited 10 times. It is a very good paper introducing a new approach to count-based exploration in deep reinforcement learning based on using binary hashcodes. The approach is interesting, the presentation is didactical, the results are good and the related literature is well covered. I learned a lot from reading this paper, so my only criticisms are on secondary aspects. I have few global points: - I think it would be interesting to first train the convolutional autoencoder, and only then use your exploration method to compare its performance to state-of-the-art methods. This would allow for a more detailed analysis by disentangling the factors of potential inefficiency. - I would put the "related work" section more forward in the paper, as the presentation of the methods would benefit from the points made in the first paragraph of related work and in the comparison with [4]. As a reader, I found that these points were important and arriving too late. - Maybe section 5.4 of supplementary material should move into the main paper, and I don't share your main conclusion about it: If you look closely at Table 6, using n(s,a) is most often better than using n(s). To me, prefering n(s) to n(s,a) based just on the max in your table is inadequate, because maybe with an "optimized" (k,\beta) pair, you will find that using n(s,a) is better... More local points: line 41: "However, we have not seen a very simple and fast method that can work across different "domains." How do you disqualify VIME here? line 113: "An alternative method must be used to ensure that distinct states are mapped to distinct binary codes." Maybe the writing could be more direc there, presenting directly the appropriate approach. Fig5: despite magnifying the picture several times, I cannot see the baseline. Didn't you forget it? Footnote 4: the statement is weak. You should give reference to the state-of-the-art performance papers. In the supplementary part: Section 1 would benefit from a big table giving all params and their values. Beyond you, it would be nice if the community could move towards such better practices. As is, Section 1 is hardly readable and it is very difficult to check if everything is given. Section 3: why do you show these images? Do you have a message about them? Either say something of interest about them, or remove them! line 136: can (be) obtained (this was the only typo I could find, congratulations! ;))

Reviewer 2



This paper presents a novel way of doing state-visit counts for exploration in deep RL. They discretize the state space with a hash function and count visits per discretized state in the hash table. Exploration reward bonuses are given based on these visit counts. The authors present three different ways of doing the hashing. The first is to use a method from the literature called SimHash, based on Locality Sensitive Hashing. The second approach is to train an autoencoder on states with a binary code layer in the middle. This binary code layer is then used as the input to the SimHash function. Finally, they use hand-crafted features from Atari (BASS) and pass these through to SimHash. The results are mixed. On continuous control tasks, they perform better than VIME on 2/4 tasks and worse than it on 2/4 tasks, using the SimHash approach. On Atari, they get the best results with the auto-encoder version on 2/6 games. In the comparison with ref 4, the authors say that approach requires designing and learning a good density model while there's only uses simple hash functions. This comparison does not seem quite fair, the author's approach that compares best to ref 4 requires designing and learning an auto-encoder to get the binary state code. The idea of discretizing continuous state into a hash table to track visit counts for exploration is a good one. However, it would be nice if there was some single approach that worked across the set of domains instead of 3 different ways of calculating visit counts. The results are also not clearly better than other approaches for doing exploration reward bonuses. Thanks for your feedback and clarifications that even your pixel based version of SimHash is performing better than state-of-the-art approaches.

Reviewer 3



Summary: This paper combines count based exploration with hashing in order to deal with large and continuous state spaces. Two different types of hashing are examined. Static hashing, and learned hashing using autoencoders. As is shown in the experimental results section, this approach leads to state-off the art performance in 2 out of the 6 tested Atari 2600 video games, and competitive performance on 1 more. Quality: There are a number of areas where this paper could improve on: - The fact that the performance of the proposed approach is more than an order of magnitude worse than the state of the art in Montezuma, and significantly worse than the state of the art in 2 more domains should not be glossed over. A discussion on the reasons that this is the case should be included in the final version of the paper. - The experiments in figures 3 and 4 were only repeated over 3-5 random seeds. This is far from enough to obtain statistically significant results. - Over how many random seeds were the experiments in table 1 that are not part of figure 4 repeated? Clarity: Overall this paper is well written and easy to understand. One thing that struck me as strange is that according to the supplementary material making the exploration bonus depend on state-actions rather than just states does not lead to better performance. It would be nice to see a discussion about why this may be the case in the main body of the paper. Originality: As far as I can tell the approach presented in this paper is original. Significance: This paper presents a significant step forward in bridging the gap between theory and practice in reinforcement learning. I expect the results presented to serve as a building block for followup work.